# Quasar: Easy Machine Learning for Biospectroscopy

**DOI:** 10.3390/cells10092300

**Published:** 2021-09-03

**Authors:** Marko Toplak, Stuart T. Read, Christophe Sandt, Ferenc Borondics

**Affiliations:** 1Faculty of Computer and Information Science, University of Ljubljana, Večna pot 113, SI-1000 Ljubljana, Slovenia; 2Canadian Light Source, Inc., 44 Innovation Boulevard, Saskatoon, SK S7N 2V3, Canada; stuart.read@lightsource.ca; 3SOLEIL Synchrotron, L’Orme des Merisiers, Saint Aubin-BP 48, CEDEX, 91192 Gif sur Yvette, France; christophe.sandt@synchrotron-soleil.fr (C.S.); ferenc.borondics@synchrotron-soleil.fr (F.B.)

**Keywords:** open source, machine learning, visual programming, data exploration, data analysis

## Abstract

Data volumes collected in many scientific fields have long exceeded the capacity of human comprehension. This is especially true in biomedical research where multiple replicates and techniques are required to conduct reliable studies. Ever-increasing data rates from new instruments compound our dependence on statistics to make sense of the numbers. The currently available data analysis tools lack user-friendliness, various capabilities or ease of access. Problem-specific software or scripts freely available in supplementary materials or research lab websites are often highly specialized, no longer functional, or simply too hard to use. Commercial software limits access and reproducibility, and is often unable to follow quickly changing, cutting-edge research demands. Finally, as machine learning techniques penetrate data analysis pipelines of the natural sciences, we see the growing demand for user-friendly and flexible tools to fuse machine learning with spectroscopy datasets. In our opinion, open-source software with strong community engagement is the way forward. To counter these problems, we develop Quasar, an open-source and user-friendly software, as a solution to these challenges. Here, we present case studies to highlight some Quasar features analyzing infrared spectroscopy data using various machine learning techniques.

## 1. Introduction

Spectroscopy offers insight into the chemical and physical properties of materials. Modern equipment can collect data at rates that make direct interpretation impossible for humans. For example, a single Fourier-transform infrared spectroscopy (FTIR) imaging microscope slide scan can generate more than a terabyte of data [1]. Such data volumes are reasonable and necessary in fields such as biomedical research where sufficient replicates are required. As a result, data analysis has become the bottleneck of scientific progress in many fields.

We approach data processing in various ways. Scientists specializing in data analysis often prefer to code, while non-programmers use various open-source and commercial software. Each approach has shortcomings, such as the difficulty of using and modifying lab-specific scripts, adapting to command-line tools, or the cost and inflexibility of closed-source environments. The use of multiple measurement techniques, common in biomedical research, results in different output data formats. If each format requires specialized software, it may be difficult to interpret the combined data.

Experiments require some degree of interactivity. Instrument manufacturers usually implement practical basic processing and visualization tools for this reason. They are especially useful for multidimensional data as inspecting a projection along a few variables (images, volumes or movies) during an experiment to help fine-tune parameters boosts data quality. However, interactivity is also crucial for data exploration, analysis, and producing final visualizations.

Complex, large volumes of data produced by the types of experiments discussed above requires a statistical approach for real understanding. Baker et al. [2] described a protocol widely used for the investigation of biological materials with infrared spectroscopy that is generalizable to most experiments. They divide data analysis into preprocessing, which addresses common issues with spectral data acquisition, and the application of multivariate statistical methods, such as clustering or supervised prediction. Some multivariate techniques, such as PCA or k-means clustering, are quite commonly used in the natural sciences, but the recent machine learning and deep learning boom of the tech industry has also found its way to the natural sciences resulting in new approaches for data analysis [3,4,5,6]. Tools that allow the application of recent machine learning methods usually require strong programming skills and until very recently user-friendly software for such studies was lacking.

In view of these issues, we see the highest potential in extendable, well-designed, user-friendly open-source software and the communities built around developing them into robust, and widely functional packages. We develop Quasar [7,8,9] as a solution to these challenges and present here applications in biospectroscopy data analysis.

## 2. Methods

Quasar is based on Orange [10,11,12,13], a user-friendly, open-source tool for machine learning and data visualization. Orange provides an accessible and interactive environment that offers a high degree of functionality while remaining adaptable to specific needs [14]. Orange offers:**Visual programming:** Orange does not impose a preset order of operations. Instead, it offers components—widgets—that either process, visualize or model inputs. Users can connect them as they see fit as long as they share connection types. This approach allows the creation of flexible workflows;**Immediate feedback:** Orange follows the general principle that being able to observe the effects of actions, and adapt immediately, improves efficiency. The option to inspect the results at every step of the analysis helps users gain confidence in results and familiarity with the analysis procedures [14];**Interactive visualizations:** Orange allows interaction with visualized elements, which can influence further analysis—for example, selecting a point on a scatter plot sends the associated data to the output. This principle empowers users to further explore interesting elements identified in the visualization;**Machine learning:** Orange includes components for unsupervised and supervised modeling and their evaluation. Mainly, it wraps established machine learning libraries for Python, such as scikit-learn or XGBoost, into user-friendly GUI components. Methods in Orange include various clustering methods, t-SNE, random forests, support vector machines, gradient boosting, and neural networks;**Extendability and modularity:** Orange is mainly written in Python, with computationally intensive parts in C [13]. It builds on Python data science libraries, such as NumPy, SciPy, Pandas, and scikit-learn. If predefined components do not suffice, the *Python Script* widget allows adding custom Python code into the workflow. Additionally, Orange provides a well-documented programming interface for adding new components and modules. More than a dozen specialized modules currently exist, including text processing, gene expression data analysis [15], image analytics [14], and time series analysis.

Quasar is a packaged distribution of Orange, extended with specific, preselected add-ons, that we provide as a single installer. Currently, it extends Orange with components specific to spectral data analysis. The spectroscopy add-on, Orange Spectroscopy, evolved from the Infrared Orange add-on [16] and shares the same basic features with a plethora of improvements and new functionality. Compared to Infrared Orange, Orange Spectroscopy adds new spectral preprocessing methods along with many user interface and general stability improvements. Since our first report in 2017 [16], we added more than 30,000 lines of code into the orange-spectroscopy GitHub repository alone. Quasar’s new components for spectral data were designed for maximum interoperability with the rest of Orange. Therefore, all machine learning capabilities of Orange are available to the spectroscopist: techniques such as clustering or classification can be directly applied to spectral data. We achieve this by embracing Orange’s *Table* data structure and conforming to its ways of data processing.

To extend the data analytics capabilities provided by Orange, Quasar currently adds tools specialized for spectroscopy in several categories (Figure 1):**Data input:** Quasar supports native data reading from major instrument manufacturers, commonly used exchange data formats and some specialized instruments;**Spectral preprocessing:** Commonly used preprocessing methods for smoothing, derivatives, baseline removal, normalization, EMSC, ME-EMSC, integration, peak fitting, etc.;**Visualization:** Plotting widgets for individual spectra, hyperspectral maps with visible image overlays, and maps of spectral series are available.

We develop Quasar as an open-source project available on GitHub under the GPL 3.0 license. Installation packages are available for major operating systems (Windows, macOS, and Linux).

## 3. Results and Discussion

This section highlights the capabilities and benefits of visual programming using Quasar through three case studies. Note that data and experiment types are not restricted to biospectroscopy; practically any type of data could be analyzed. The workflows we present include all crucial analysis steps recommended by Baker et al. [2].

### 3.1. Case Study 1: Identification and Localization of the Medullas in Human Hair Sections

To showcase unsupervised learning on spectral data with Quasar, we will use an FTIR hyperspectral dataset of multiple hair sections measured in transmission mode as described in detail in the study of Sandt and Borondics [17]. The dataset contains absorbance values in the 4000–800 cm^−1^ range. Figure 2 shows how Quasar allows identification of interesting groups of spectra: in this case, detecting the hair medulla. In the workflow, we used unsupervised learning—clustering—twice: first to remove physical variations such as the background or scattered edges, and then, to detect variations in the chemical composition inside the hair sections. Typically, we would use thresholds according to the intensity (the *Select Rows* widget in Quasar) to remove the background. On this dataset, *k-Means* clustering provided the separation we required.

To achieve a useful identification of groups of similar spectra, we clustered second derivatives of original spectra. Therefore, the output—spectral graph in the last *HyperSpectra* widget in the Figure 2c—is shown in the derivative space. Spectra in the derivative space are often harder to interpret. Thus, to investigate the selected spectrum of the medulla in the space of original data, we used a *Spectra* with the selected spectrum as its subset input; there, we were able to observe the selected spectrum before any processing. The spectrum reveals a characteristic lipid signature. Even though spectra in the dataset are saturated in the Amide I and II regions, the methods we employed were robust enough to identify a known biological structure, i.e., the medulla.

The case study showcases how immediate feedback influences the analysis. If we did not preprocess the data before the second *k-Means* clustering, the clusters obtained are not meaningful. On these data, seeing clustered display (in *Hyperspectra*) while searching for appropriate preprocessing proved invaluable: an interesting clustering result steered the analysis in a particular direction.

We used k-means clustering to find groups of similar spectra. Straightforward alternatives for grouping, such as hierarchical clustering or DBSCAN, would likely yield similar results, but k-means was computationally the most efficient. Furthermore, dimensionality reduction techniques such as principal component analysis, multidimensional scaling (MDS), or t-distributed stochastic neighbor embedding (t-SNE) applied before grouping could help better understand the structure within groups. Quasar offers all listed techniques.

One of the design principles of Quasar is that anything displayed can be analyzed further. In the workflow shown in Figure 3, we applied the principal component analysis (PCA) to clusters obtained in Figure 2. We needed to apply PCA to the original spectra, not the derivatives used for clustering, to understand the cluster’s spectral characteristics. We thus merged the resulting clusters with the original data and, after preprocessing, applied PCA. In Figure 3c, a *Scatter Plot* shows that the first principal component easily identifies the cluster, and the *Spectra* shows the corresponding loadings with the lipid signature.

### 3.2. Case Study 2: Building a Drug Resistance Prediction Model

Here, we apply supervised learning on FTIR spectra obtained from three different cell types, aiming to develop a statistical model for predicting resistance to all tyrosine kinase inhibitors (drugs aimed for treating chronic myeloid leukemia) conferred by the T315I mutation in the *BCR-ABL* gene [18]. The spectra in the dataset correspond to individual cells measured with synchrotron radiation FTIR microspectroscopy in transflection mode, each described by absorbance values in the 1800–900 cm^−1^ range. Figure 4 shows the workflow and its most important components. After preprocessing, we evaluated two supervised prediction methods, logistic regression and random forests, using five-fold cross-validation. Looking at the *Test and Score* widget, it is clear that random forests performed better for all calculated metrics.

Next, we closely investigated the prediction by random forests. The *Confusion Matrix* widget shows that random forests perfectly predicts the “Resistant” class, but sometimes mixes the “Wild-type” and “Bcr-abl” classes. Here, further analysis could show why these two classes are similar; if we selected mispredicted classes in the *Confusion Matrix*, they would be sent to the output, where we could analyze them further. To interpret the random forest model, we ranked wavenumbers according to their importance and displayed these ranks as *Spectra*. We can conclude that the region from 1600–1500 cm^−1^ is the most important for prediction. This region partially represents the protein spectral signatures. Interestingly, the wavenumber regions at the edges have high importance, which might be due an artifact.

Now that we have established the Quasar workflow, we could modify its settings and see the downstream effects. For example, changing preprocessing would influence the evaluation results, mispredictions, and the importance of features. Such exploratory modifications and observing their results can help gain valuable insight into the data.

Quasar implements current general-purpose supervised learning techniques, such as support vector machines, neural networks, and gradient boosting, which could have been used in this case study. Other modifications to the workflow, such as dimensionality reduction before model induction, could also benefit the analysis. Additionally, we could have inspected the obtained random forests further by looking into individual trees. The chosen models could also be saved and reused later to classify new measurements.

### 3.3. Case Study 3: Protein Secondary Structure Peak Fitting

Next, we highlight how peak fitting capabilities [19] are integrated into a larger workflow in Quasar using spectra from a study on human cirrhotic liver tissue [20] measured with FTIR and annotated according to the majority presence of a chemical compound (collagen, glycogen, lipids, or DNA) in that part of the cell. In a recent paper, Stani et al. [21] established a correlation between two distinct protein spectral regions (Amide I and Amide III) for collagen during thermal degradation. In their analysis, they use spectral preprocessing and peak fitting to extract secondary structure characteristics. Figure 5 shows the workflow which implements their described peak fitting method in Quasar.

After filtering the dataset to select rows labelled as “collagen”, we split the spectra into two sub-regions, Amide I (1750–1580 cm^−1^) and Amide III (1310–1175 cm^−1^), and identically preprocessed them with rubber band baseline correction and min-max normalization. We configured peak fitting, bands, components, and fitting constraints as described in [21], and obtained starting parameters from an initial fit performed on the average spectrum for the final full-dataset fit. The *Peak Fit* widget (Figure 5a; Amide III) shows the resulting fits for some example spectra, while fit result parameters, total fits, and residuals are output for the entire dataset.

Next, we can analyze the resulting fit parameters as any other feature in Quasar. Stani et al. [21] found a positive correlation between the Amide I 1690 cm^−1^ and the Amide III 1284 cm^−1^ components, assigning both to the carbonyls which comprise an intra-strand hydrogen bond network stabilizing the triple-helix fibrils. Despite the lack of perturbation in the dataset used here, we also observe a positive correlation between these component areas (Figure 5b) within the natural variation of the protein. We also find a correlation between the 1660 and 1240 cm^−1^ components, assigned to the well-known inter-strand hydrogen bonding in the triple-helix.

As the fit results are integrated into the Quasar workflow they can be further explored using approaches similar to the previous case studies.

### 3.4. Limitations and Further Development

Quasar, similarly to most general-purpose scientific software such as R, Matlab, or SciPy, requires the complete dataset in memory. Big data require specialized libraries that, instead of loading the entire dataset into memory, analyze the data while passing through it directly on a disk. The authors of the memory-efficient SIproc library [1] report that the rate of reading through the data is commonly their libraries’ speed-limiting factor. Still, their library only provides a programmatic interface excluding non-experts. Following the same philosophy, we are currently changing Orange and Quasar extensions to support out-of-memory data handling.

### 3.5. Current Project Reach

Currently, we know of 35 published scientific publications that successfully used Quasar or the Orange Spectroscopy add-on for data analysis (an up-to-date list is available at https://quasar.codes/publications/, accessed on 1 September 2021). Workshops about spectral data analysis with Quasar took place in Italy, Slovenia, Norway, France, Belarus, Canada, Australia, and, since COVID-19, online with participants from the UK, Sweden, Germany, and Egypt.

## 4. Conclusions

Here, we introduced Quasar, a software based on Orange, to reform data analysis in the natural sciences. Currently, Quasar provides tools and functionality aimed at spectroscopy data analysis and combines them with powerful machine learning methods. Thanks to the visual programming approach of Orange, scientists without any coding background can build flexible data processing workflows and explore various analysis ideas interactively. With the expansion of the Quasar community, and contributions from expert developers to the open-source code, functionality can be extended towards new fields, measurement and data analysis techniques. The free availability of Quasar addresses bottlenecks in adoption. In conclusion, we strongly believe that this approach will empower scientists to better and easier understanding of experimental data.

## Figures and Tables

**Figure 1 cells-10-02300-f001:**
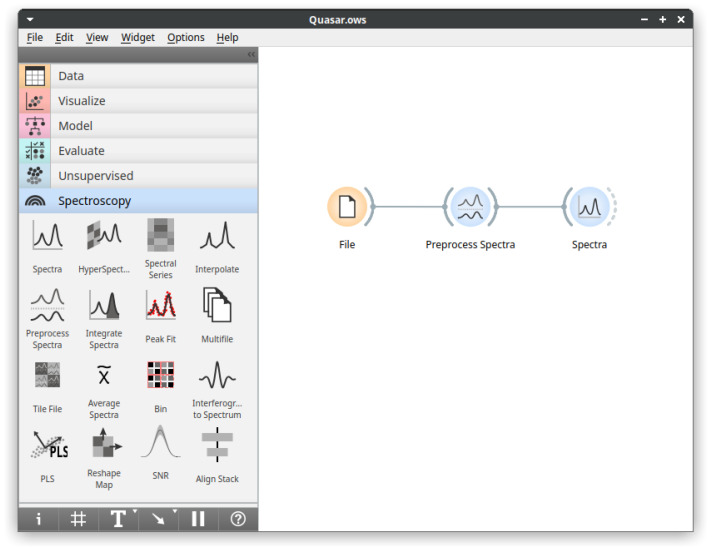
The main Quasar window. The left part of the window contains available analysis components (widgets) organized into categories with the Spectroscopy category open. The right part of the widow is the workflow editor; the displayed workflow performs basic data inspection.

**Figure 2 cells-10-02300-f002:**
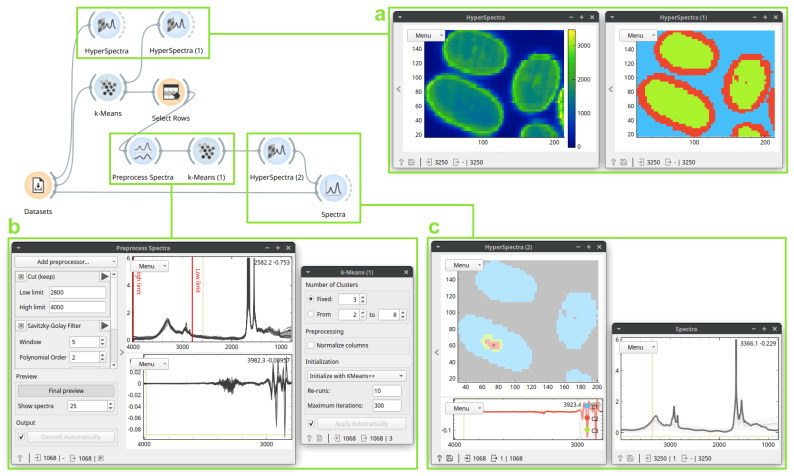
An unsupervised analysis on a dataset of multiple hair sections. The *HyperSpectra* (box **a**, **left**) displays integrals of the spectral content. The first *k-Means* clustering outputs three groups of spectra based on their spectral similarity. As these groups correspond to sample morphology well (*HyperSpectra*, box **a**, **right**), selecting (*Select Rows*) only the innermost removes both the background and the edges (where scattering occurs). After preprocessing (a wavenumber region selection and second derivatives; *Preprocess Spectra*, box **b**), *k-Means* clustering discovers a cluster at the center of the largest cross-section (*HyperSpectra*, box **c**, **left**). A selected spectrum from the cluster in non-derivative space, shown in the *Spectra* widget (box **c**, **right**), contains a lipid signature characterized by peaks between 3000–2800 cm^−1^.

**Figure 3 cells-10-02300-f003:**
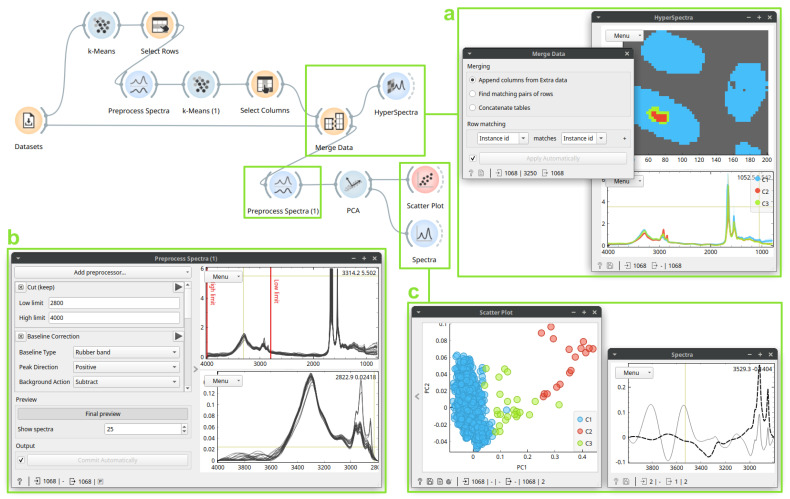
An unsupervised analysis on a section of multiple hair samples, continued from Figure 2. Here, a *Merge Data* (box **a**, **left**) widget merges the clusters obtained with the previous data for further analysis. The clusters merged with original spectra may be seen in the *Hyperspectra* widget (box **a**, **right**). Here, *Preprocess Spectra* baseline-corrects and normalizes the data (box **b**). We then investigate results of the *PCA*: in the *Scatter Plot* (box **c**, **left**), the red cluster has high PC1 scores, whose loadings, shown in the *Spectra* (box **c**, **right**), contain a characteristic lipid signature between 3000–2800 cm^−1^.

**Figure 4 cells-10-02300-f004:**
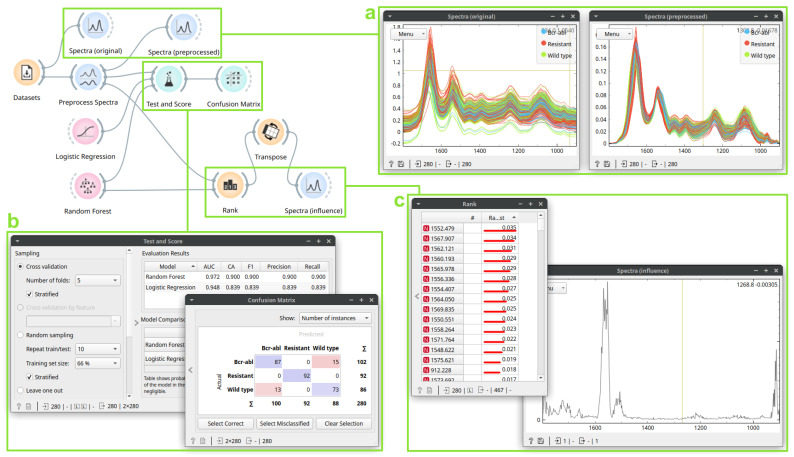
A workflow for supervised analysis of three cell lines. The first *Spectra* (box **a**, **left**) displays the original data, colored by the cell line. The second *Spectra* (box **a**, **right**) shows data after preprocessing: *Preprocess Spectra* applies rubber band baseline and vector normalization. The *Test and Score* (box **b**, **left**) widget shows cross-validated prediction quality of two methods: *Logistic Regression* and *Random Forest*. The *Confusion Matrix* (box **b**, **right**) widget highlights errors from *Random Forests*: for the “Resistant” class, there were no errors. Finally, the *Rank* (box **c**, **left**) shows wavenumber importance for the random forest classifier; these are also shown in the *Spectra* (box **c**, **right**) widget: the region 1600–1500 cm^−1^ is the most important for classification.

**Figure 5 cells-10-02300-f005:**
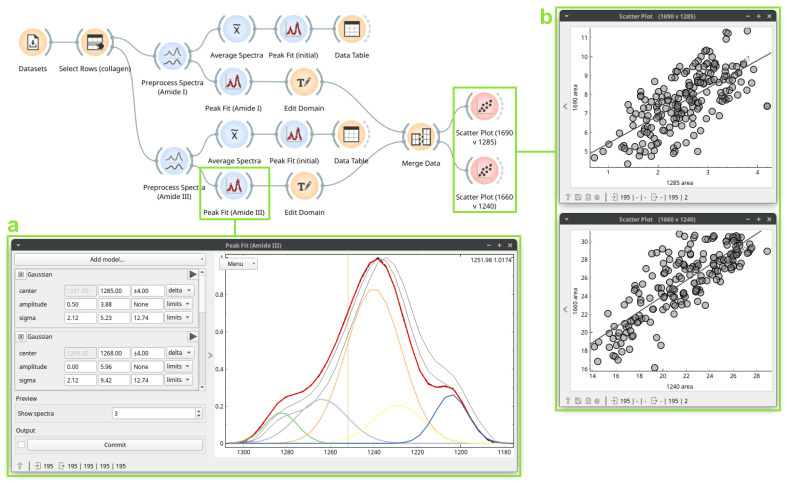
A workflow for protein secondary structure peak fitting. The *Preprocess Spectra* cuts the data to the Amide I or Amide III region, applies rubber band baseline and min-max normalization. The *Average Spectra* computes the average spectrum. The first *Peak Fit (initial* fits the average spectrum with constraints and default initial values. The second *Peak Fit* (box **a**) computes the fit for each spectrum, using initial amplitude and sigma values from the average fit. The *Edit Domain* labels the peak area features by center wavenumber. The *Merge Data* combines the Amide I and Amide III peak fit datasets. Finally, the *Scatter Plot* widgets (box **b**) show the relationship between selected peak areas for the entire dataset.

## Data Availability

All data presented in the case studies are freely available from the *Datasets* widget in Quasar. To replicate, modify and explore our case studies, the workflow files can be found in a GitHub repository: https://github.com/Quasars/supp-cells-2021, accessed on 1 September 2021.

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
