# Peer review of "Quasar: Easy Machine Learning for Biospectroscopy"

_cells, 2021, doi:10.3390/cells10092300_

Round 1

Reviewer 1 Report

In this article the authors describe their development of an open source spectroscopic data analysis tool. The tool was developed to be hands on and easy to utilize, even for a non-computer-scientist. The tool implements a number of algorithms with visualization and data manupulation techniques able to be implemented at each step. 

This type of data analysis software, especially in spectroscopy, is incredibly valuable and many companies charge expensive fees, close the source, allow no modification to the software, and don't provide the workflow-based methods presented by the authors. This is a valuable addition to the field of spectroscopy in the relatively new field of "big data spectoscopy."

The authors utilize this manuscript to demostrate the usefulness of this software through three interesting cases. The scientific validity of an conclusions is not simple to ascertain through a demonstrative example, but the science underlying the spectoscopic data in this manuscript isn't really the point. The demos are worthwhile and clearly demonstrate the usefullness of the analysis and analysis methods within their software. 

It is essential to advancing this field that such analysis tools are available and this tool will prove useful to biospectroscopists. Though the manuscript is more of an application of the analysis tool than a scientific study, it is clearly written, valuable, and worthy of publication.

I reccommend accepting the manuscript as is.

Reviewer 2 Report

In this paper, the authors successfully developed an open-source and user-friendly program called Quasar, which is proven to be useful for spectroscopy data analysis.  Overall, the manuscript is written comprehensively with well-presented data and fathomable for broad readers. This machine learning approach is of great alternative for big data analysts in multidisciplinary areas, especially with its easy user interface system. It is then suitable for being published in Cells MDPI, particularly in the section for Biophysics. However, several issues should be addressed to improve the quality of this work before its publication. Here are the comments raised by the Reviewer:

  1. Three case studies have been investigated with Quasar, but the specific data collection process is not clearly explained in the method section. Could the authors briefly add an explanation of how the data for each case were obtained?
  2. In each case study, the machine learning techniques that have been used are already determined (i.e., the clustering method is chosen in case study 1). Meanwhile, there are many types of machine learning methods so far. Dealing with this issue, does the Quasar provide the alternative methods that are applicable for any specific case study?
  3. The authors mentioned that Quasar software is user-friendly, especially to both experts and non-experts of visual programming. If possible, could the authors provide such supplementary data showing the percentage of the user that successfully applied the Quasar program for their experimental data analysis? In doing so, this would be powerful to emphasize the relevance of Quasar software for worldwide scientists with various research fields.
  4. In this work, the Quasar functionality is narrowed to the bio-spectroscopic data. Considering many types of bioanalytical techniques, is there any possibility to expand this software function to different types of datasets? If yes, it would be a great vision that can be simply explained with a concrete sample in the conclusion part.
  5. The sentence in line 28-30 is illegible. Please try to rephrase it.
  6. The sentence in line 166-167 is better to be separated into two clauses.

Round 2

Reviewer 2 Report

The manuscript is well-revised and thus is suitable for publications.